# Identifying core leadership competencies to success non-communicable disease control and prevention programs: A mixed-methods study

Jafar Sadegh Tabrizi[1], Yegane Partovi[2*], Andrew Wilson[3], Kamal Gholipour[4], Mostafa Farahbakhsh[5], Tohid Jafari Koshki[6], Ahmad Koosha[7], Jabreil Sharbafi[8]

**1** Health Services Management, Tabriz Health Services Management Research Center, School of Health Management and Medical Information, Tabriz University of Medical Sciences, Tabriz, Iran, **2** Health Services Management, Department of Public Health, School of Public Health, Ardabil University of Medical Sciences, Ardabil, Iran, **3** Public Health, Menzies Centre for Health Policy and Economics, School of Public Health, the University of Sydney, Sydney-Australia, **4** Health Services Management, Health Services Management Research Center, School of Management and Medical Informatics, Tabriz University of Medical Sciences, Tabriz, Iran, **5** Psychiatrist, Research Center of Psychiatry and Behavioral Sciences, Tabriz University of Medical Sciences, Tabriz, Iran, **6** Biostatistics, Molecular Medicine Research Center, Department of Statistics and Epidemiology, Faculty of Health, Tabriz University of Medical Sciences, Tabriz, Iran, **7** Public Health, National Public Health Management Center, School of Public Health, Tabriz University of Medical Sciences, Tabriz, Iran, **8** East Azerbaijan Provincial Health Centre, NCD Department, Tabriz, Iran

* Modirpy@yahoo.com

## Abstract

### Introduction

Strengthening strong leadership skills is essential for program managers to successfully implement programs for the prevention and control of noncommunicable diseases (NCDs). This study identified leadership competencies for managers of individuals with noncommunicable diseases in Iran.

### Methods

The study had three steps: a literature review, an expert panel, and a Delphi technique survey. First, the literature was reviewed to compile a list of leadership competencies in the field of primary health care (public health, NCDs). To refine and adapt the original list of leadership competencies, it was provided to the expert panel in two stages. The list of leadership competencies was sent via email to 30 experts over the course of two Delphi rounds. Descriptive statistics were used to analyze the outcomes.

### Results

Fifteen leadership competencies, comprising multisectoral collaboration, political awareness, evidence-informed decision making, risk and disaster management, planning, innovation, leading and managing change, team building, communication, quality

**Data availability statement:** All relevant data are within the manuscript and its Supporting Information files.

**Funding:** The author(s) received no specific funding for this work.

**Competing interests:** The authors have declared that no competing interests exist.

**Abbreviations:** NCDs, Noncommunicable diseases; PHC, Primary health care.

improvement, systematic thinking, management, ethics and professionalism, motivation and inspiration and personality, were identified.

## Conclusion

The leadership competencies identified in this study can be a helpful tool in evaluating and identifying skills, knowledge, and attitudes with program managers for the prevention and control of NCDs and in designing training programs to strengthen leadership skills.

## Introduction

The leading causes of death worldwide are NCDs (Noncommunicable diseases) such as cancer, osteoporosis, allergies, diabetes, chronic lung disease, and cardiovascular diseases. In 2019, statistics indicate that NCDs caused 74.4% of all deaths and that 63.8% of years spent with a disability [1]. According to a report by the World Bank, mortality from NCDs will increase from 28% in 2008 to 46% by 2030. As a result, the World Economic Forum and Harvard University estimate that NCDs will cost the world economy $47 trillion over the next 20 years, which is equal to 75% of the global gross domestic product [2]. NCDs also reduce productivity, which lowers capital formation and slows economic growth [3]. NCDs are more prevalent in countries with the poorest and most deprived populations (at all levels of development). The evidence generally demonstrates that NCD-related mortality causes a disproportionate burden on the poor and is a threat to social and economic growth. A staggering 100 million people fall into poverty each year, nearly 17 million of whom reside in the Eastern Mediterranean Region. This is attributed to the high direct costs of NCD care and the loss of productivity through disability and death of the household head [4]. The decline in premature mortality from NCD worldwide was 15% between 2000 and 2012. The rate of decline is not enough to meet Target 3.4 of the Sustainable Development Goals, which is to reduce premature mortality from NCD by one-third by 2030 [5].

Given the multidimensional nature of NCDs, the health system must implement complex interventions that require substantial financial resources, skilled and trained workforce resources, coordinated cooperation between various social organizations, political commitments, equipment, and information constructions [6,7]. These capacities are essential prerequisites for implementing NCD programs but crucially require managers with strong leadership skills who can coordinate, support, and ensure their long-term continuity to produce beneficial outcomes [8]. To ensure that effective interventions are delivered efficiently within available resources, especially in low- and middle-income countries, leaders with the required competencies and qualifications are essential [6,9,10].

The ability to carry out a duty and activities that are connected to the job description is referred to as competency. Competency is also a set of knowledge, attitudes and skills needed to perform a job [11–13]. The World Health Organization has placed an emphasis on enhancing leadership and management competencies and skills at the level of public health programs [14]. Research indicates that the greatest obstacle to providing effective services is the lack of managers with recognized managerial and leadership competencies and skills at all levels of the health system [15]. Experience with the implementation of many programs in Iran demonstrates challenges related to resourcing, failure to appropriately consider existing infrastructure, a lack of clarity in the definition of duties and responsibilities and their incompatibility with the activities requested in NCD programs, and inadequate support for primary care program managers [16]. The appointment of primary healthcare managers without regard to leadership qualifications and management training relevant to public health programs is a

significant factor in the root cause analysis of inadequate program implementation. Notably, while the majority of newly appointed managers are medical and paramedical graduates, they are usually completed in management and leadership courses [15].

Therefore, it is essential to identify strong leadership competencies for managers of programs for the prevention and control of NCDs to overcome the obstacles associated with NCD programs. The current education programs for PHC managers worldwide do not provide clear leadership competencies for NCD program management [17,18]. This study identified leadership competencies for managers of noncommunicable diseases in Iran for the first time.

## Methods

A mixed-method approach was used, including a literature review, an expert panel, and a Delphi technique survey. A summary of the methods is presented in Fig 2. All parts of the expert panel were performed according to the guidelines provided in the COREQ (COnsolidated criteria for REporting Qualitative Research) checklist (Refer to attachment 1) [19]. This study was conducted between January 2022 and the end of April 2023.

### Participants and recruitment

Purposive sampling was used to recruit participants; that is, participants who were particularly knowledgeable or experienced and ready to convey experiences and opinions related to the study aim were identified and selected. The participants included officials and headquarters staff of NCDs at the Iranian Ministry of Health and Medical Education (MOHME), the head of the health service management research center, faculty members in the fields of public health, health services management, social medical, and provincial-level NCD department managers from all universities affiliated with the Iranian MOHME. The inclusion criteria consisted of having at least three years of management experience or executive and scientific activities in the fields of PHC and NCDs and having the qualifications and willingness to attend the interviews. The exclusion criteria included unwillingness and lack of interest in participating in the study. Written consent was obtained from all the expert panel and Delphi participants, and they were told that they were not obligated to participate in the research and could leave the interview whenever they wanted. This study is based on a doctoral thesis approved by the Research Ethics Committee of the Tabriz University of Medical Sciences (approval code: IR.TBZMED.REC.1399.167, Phazhohan code: 63953).

### Procedure

1. **Generation of an initial list of leadership competencies.**

- **Literature review:** The electronic databases PubMed, EMBASE, PROQUEST, Web of Science, and SCOPUS were searched via the keywords "leadership characteristics", "chronic disease management", "noncommunicable disease management", "task", "skill", and "professional competencies" for the competencies necessary for leadership in the field of PHC (public health, noncommunicable diseases) (see Table 1). The MOHME website, the websites of medical sciences universities, the repository of academic theses, management and policy magazines, health assessments, reference studies, and the Google site were also manually searched to extract gray literature. Additional research was conducted on rules and guidelines, higher laws, job descriptions, and responsibilities. The inclusion criterion was that the full text should be published by the end of 2022 in Persian or English and focused on leadership competencies. Abstracts of articles published at conferences, articles that were in the management and leadership field but not in the field of health, and papers focusing solely on

Table 1. The search strategy.

| Data-base | Results | Search strategy |
|---|---|---|
| Web of Science | 80 | TOPIC: (chronic disease* OR noncommunicable OR "noncommunicable")) AND TOPIC: (leader* OR manage*) AND TOPIC: (competence OR skill OR task) Refined by: LANGUAGES: (ENGLISH) TOPIC: ("care, primary health" OR "health care, primary" OR "primary health care" OR "health care, primary" OR "primary care" OR "care, primary") AND TOPIC: (leader* OR manage*) AND TOPIC: (competence OR skill OR task) Refined by: LANGUAGES: (ENGLISH) |
| Scopus | 120 | TITLE-ABS ("skill" OR "competence" OR "task") AND TITLE-ABS ("care, primary health" OR "health care, primary" OR "primary health care" OR "health care, primary" OR "primary care" OR "care, primary") AND TITLE-ABS (lead* OR manage*) AND TITLE-ABS-KEY (noncommunicabl* OR "noninfectious disease*") |
| Embase | 79 | ("skill" OR "competence" OR "task") AND ("care, primary health" OR "health care, primary" OR "primary health care" OR "health care, primary" OR "primary care" OR "care, primary") AND (lead* OR manage*) AND ("noncommunicable chronic disease*" OR "noninfectious disease*" OR "noncommunicable disease"/exp) |
| PubMed | 77 | ((lead*[Title] OR manage*[Title]) AND ("chronic disease*" [Title/Abstract] OR "non communicable" [Title/Abstract] OR noncommunicable [Title/Abstract]) AND (skill [Title/Abstract] OR skills [Title] OR competence [Title] OR competencies [Title] OR competency [Title] OR task [Title])) |
| Proquest | 10 | Noft (chronic disease* OR noncommunicable OR "noncommunicable")) AND noft (lead* OR manage*) AND noft ("skill" OR "competence" OR "task") |

management competencies were excluded. The searches identified from databases (366 studies) and gray literature (45 records) and duplicate records were removed, and then the article titles and abstracts were independently reviewed by two researchers. From the final selection of 19 papers, a detailed review identified 150 competencies (see Fig 1). These were reviewed by researchers, and revisions were made regarding the removal or integration of specific competencies.

**2. Revision of initial leadership competencies organized into four domains.**

- **Expert panel:** Two in-person expert panel meetings (each lasting 60 minutes) were conducted to evaluate, refine, and modify the initial draft of leadership competencies. A total of 150 leadership competencies, initially identified through a comprehensive literature review, were presented to a panel of 10 experts. The panel was tasked with reviewing the competencies for appropriateness and specificity and providing feedback on their relevance. This process involved merging, eliminating, and refining the competencies on the basis of expert judgment.

During these discussions, it became apparent that organizing the competencies into thematic domains would enhance their clarity and applicability. Drawing from established competency models (such as the Bartram 8G Model), the competencies were classified into four overarching domains: specific, distinct, managerial, and behavioral. By the conclusion of the panel sessions, the number of competencies was reduced from 150 to 19, which were organized into the following domains:

- Specific: This includes the technical and specialized competencies required and crucial for directing NCD control and prevention programs.

- Distinct: This includes only the competencies that determine effective and successful leadership, where most skills are considered.

- Management: This includes all the general management abilities necessary for the management position of the NCD department.

- Behavioral: This includes all of the personality and unique characteristics of the leader (see Fig 2).

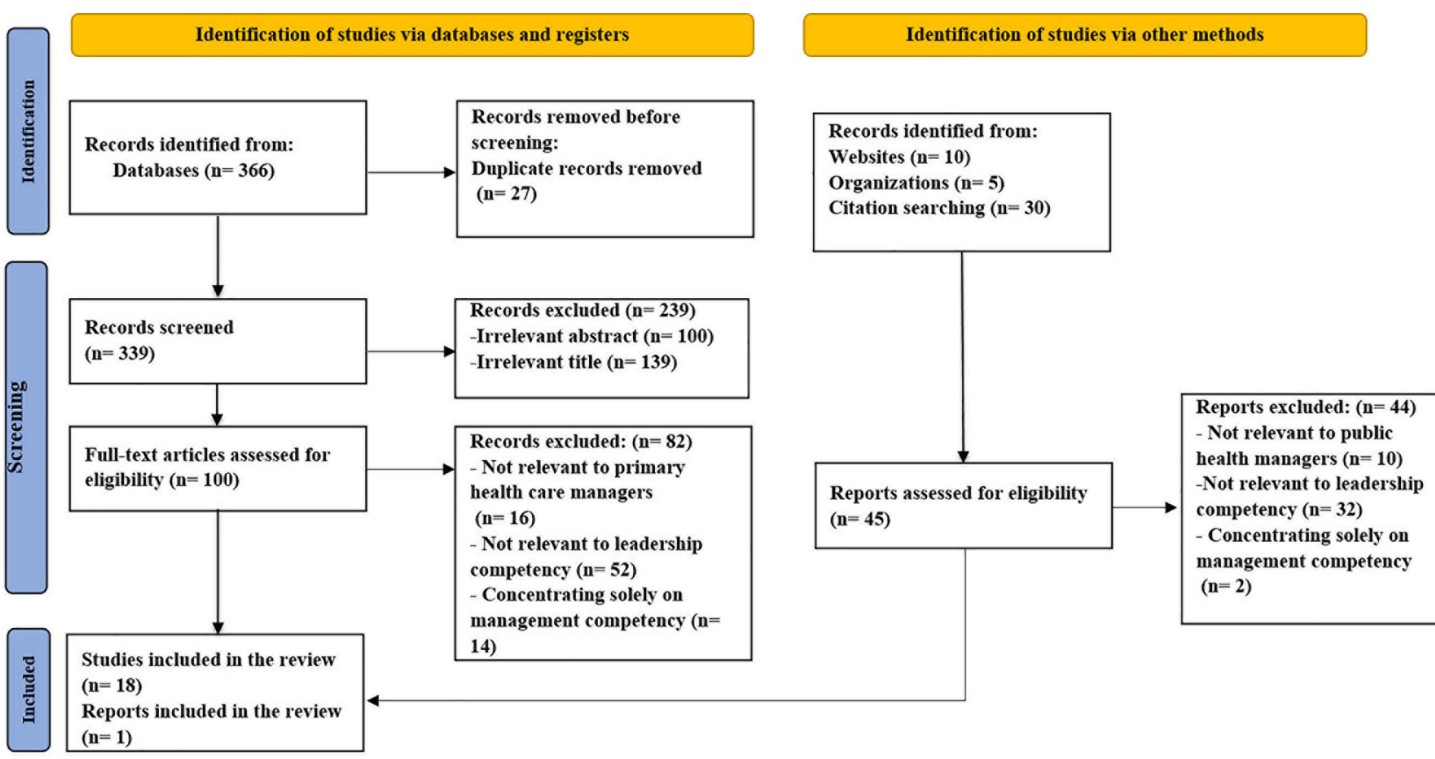

**Fig 1. Flow diagram of the study selection process.**

**3. Attaining agreement and a degree of consensus.**

- **Delphi technique survey:**    The refined competencies and their associated domains in the expert panels were further validated through a Delphi process, which was conducted in two rounds. The Delphi technique requires the cooperation of at least 10 experts [20]. The Delphi method involves 30 invited experts, 27 of whom participated in the first round and 20 of whom participated in the second round. The purpose of the Delphi process was to achieve consensus on competencies by soliciting expert opinions through a structured survey.

The Delphi process included questions about participant demographics and leadership competencies. Each competency was rated on a 9-point Likert scale on the basis of two criteria:

- Necessity: The degree to which a competency is essential for enhancing the leadership ability of program managers in the prevention and control of NCDs.

- Applicability: The feasibility of implementing the competency in the context of NCD management in Iran.

The ratings ranged from 1 (least necessary/applicable) to 9 (most necessary/applicable), with 5 representing neutrality. Three distinct ranges were defined:

- Disagreement (1–3)

- Neutrality (4–6)

- Agreement (7–9)

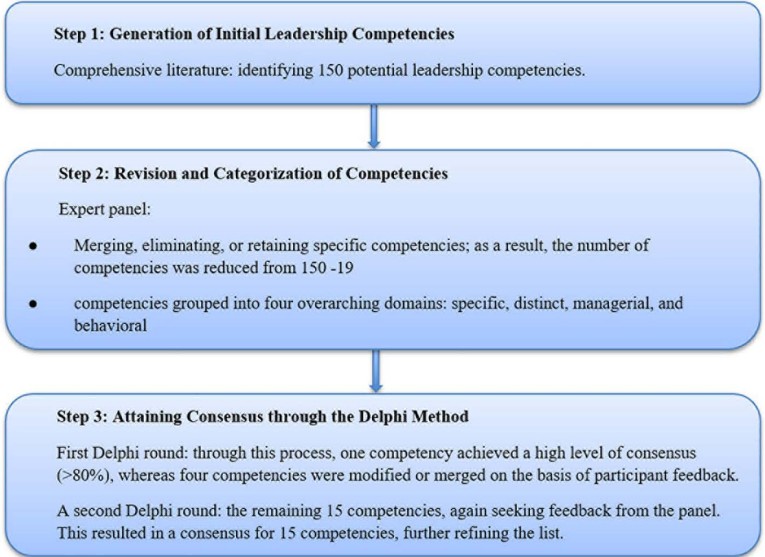

**Fig 2. Step diagram of the process of identifying core leadership competencies.**

If the median score for a competency was between 1 and 3, it was excluded from further consideration. A score between 4 and 6 indicated that the competency would proceed to the next Delphi round, whereas a score of 7 or above resulted in the direct inclusion of the competency in the final list. Additionally, participants were invited to provide qualitative feedback and suggest any new competencies or modifications.

According to the principles of the Delphi technique, a consensus was reached on the basis of the percent agreement criterion. A consensus was reached when 80% of the participants rated a competency above 7 in each round [21]. The stability between Delphi scoring rounds was assessed via the Wilcoxon matched-pairs signed-rank test. A statement was considered stable if there was no statistically significant change in response between the scoring rounds for each statement (p ≥ 0.05). The data were analyzed via SPSS 24 (see Fig 2).

## Results

Table 2 displays the results in relation to the demographic factors of the study participants.

On the basis of a literature review, 150 leadership competencies were identified and listed (see Figure 1 and Table 3).

On the basis of the expert panel input, the competencies were categorized into four domains: specific, distinct, managerial and behavioral. These groupings were as follows:

- Specific Domain: multisectoral collaboration, political awareness, evidence-informed decision making, risk and disaster management, and planning competencies.

- Distinct Domain: driving innovation, managing change, team building, communication, quality improvement, and systematic thinking competencies.

- Managerial Domain: resources, executive and administrative management competencies.

- Behavioral Domain: ethics and professionalism, motivation and inspiration, and personality competencies.

**Table 2. Participants' demographic characteristics as n (%).**

| Variables | | Frequency (%) | | |
| --- | --- | --- | --- | --- |
| | | Expert panel | Delphi round | |
| | | | First | Second |
| Gender | Male | 7(70) | 20(74.07) | 15(75) |
| | Female | 3(30) | 7(25.9) | 5(25) |
| Job experience | 20≤ | 2(20) | 10(37.03) | 7(35) |
| | 21-25 | 7(70) | 12(44.4) | 8(40) |
| | >25 | 1(10) | 5(18.5) | 5(25) |
| Education Attainment | BSc. | – | 5(18.5) | 4(20) |
| | MSc. | 1(10) | 9(33.3) | 8(40) |
| | PhD.MD | 9(90) | 13(48.1) | 8(40) |
| Occupation | Physician | 6(60) | 10(37.03) | 9(45) |
| | Non-Physician | 4(40) | 17(62.9) | 11(55) |

The 19 leadership competencies entered the first round of the Delphi phase, with nearly 90% of the invited participants responding. The titles of the 4 domains were agreed upon by all the participants. The "risk management" competency was merged with the "disaster management" competency, and the competencies of "resource, executive and administrative management" were merged into "management". The "managing change" competency was rephrased into "leading and managing change". The competencies that received 80–100% agreement in the first round of the Delphi process were accepted without further testing, whereas those that received less than 80% agreement underwent revision on the basis of participant comments. Four domains and fifteen leadership competencies were presented for the second round of Delphi after modification and editing. However, there was no disagreement among the four fields. We concluded that saturation and consensus had been reached because no additional competencies were proposed in the second round (Table 4).

On the basis of the Delphi results, the competencies were categorized into four domains: specific, distinct, managerial and behavioral. The last column of Table 5 contains specific and more detailed descriptions of each competency.

## Discussion

This study identified a set of leadership competencies for program managers in the prevention and control of NCDs in Iran. The results led to the compilation of fifteen leadership competencies into four domains titled "specific", "distinct", "managerial" and "behavioral". The consistency of the experts' responses and the high consensus scores for all the domains and competencies demonstrate acceptance by a group of health professionals who are knowledgeable about the Iranian health system.

### Specific domain

The specific domain includes multisectoral collaboration, political awareness, evidence-informed decision making, risk and disaster management, planning competencies, and relevant statements. Previous studies conducted in Iran have revealed that managers of public health programs for preventing and managing NCDs have significant weaknesses. In addition, the responsibilities and roles of organizations should be defined to gain support in identifying areas where interdisciplinary interventions are needed for NCD programs [16]. Moreover, managers in the primary care sector struggle to use information effectively. Some

**Table 3. The findings of included studies in the review.**

| Authors | Title of studies | Year | Country | Design | Competencies |
|---|---|---|---|---|---|
| Mohd-Shamsudin [22] | Determinants of managerial competencies for primary care managers in Southern Thailand | 2012 | Thailand | Mixed | • Make a vision<br>• Sense of mission and Partnership<br>• Communication |
| Katarzyna Czabanowska [23] | In search for a public health leadership competency framework to support leadership curriculum a consensus study | 2014 | European Region | Mixed | • Systems thinking<br>• Political leadership<br>• Collaborative leadership<br>• Communication<br>• Leading change<br>• Organizational learning and development<br>• Ethics and professionalism |
| Keerati Kitreerawutiwong [24] | Development of the competency scale for primary care managers in Thailand: Scale development | 2015 | Thailand | Mixed | • Clarify vision, mission, and goal<br>• Understand policy and communicate the policy to staff<br>• Innovation<br>• Emotional intelligence<br>• Strategic decision making |
| Zhanming Liang [25] | Competency requirements for middle and senior managers in community health services | 2018 | Australia | Mixed | • Evidence-informed decision-making<br>• Leading and managing change<br>• Political awareness<br>• Professionalism<br>• Relationship management<br>• Public, industrial relations and networking |
| Zhanming Liang [26] | An evidence-based approach to understanding the competency development needs of the health service management workforce in Australia | 2018 | Australia | Mixed | • Evidence-informed decision-making<br>• Ethics and professionalism<br>• Communication |
| Kinley Dorji [27] | Leadership and management competencies required for Bhutanese primary health care managers in reforming the district health system | 2019 | Bhutanese | Quantitative | • Professionalism<br>• Communication<br>• Leading change<br>• Analytical thinking<br>Innovative thinking<br>• Transformation<br>• Develop and communicate vision<br>• Promote work environment<br>• Decision-making<br>• Accountability<br>• Develop interpersonal relationship<br>• Commitment to develop others |
| Sudip Bhandar [28] | Identifying core competencies for practicing public health professionals: results from a Delphi exercise in Uttar Pradesh, India | 2020 | India | Mixed | • Visionary leadership<br>• Develop key values<br>• Organizational learning<br>• Maintaining organizational Performance standards<br>• Communication<br>• Leading change |
| Mary E Stefl [29] | Common competencies for all health care managers: the health care leadership alliance model | 2008 | American | Quantitative | • Communication<br>• Relationship management<br>• Professionalism<br>• Marketing |
| Zahra Ladhani [30] | Competencies for undergraduate community-based education for the health professions a systematic review | 2015 | Pakistan | Mixed | • Teambuilding<br>• Partnership<br>• Negotiation<br>• Leading change |

*(Continued)*

**Table 3.** (Continued)

| Authors | Title of studies | Year | Country | Design | Competencies |
|---|---|---|---|---|---|
| Raymond Lucas [31] | Leadership development programs at academic health centers: results of a national survey | 2018 | Wash-ington | Mixed | • Organizational direction<br>• Leading change<br>• Creating incentives and rewards<br>• Decision making<br>• Working with others<br>• Communication<br>• Teambuilding<br>• Marketing<br>• Emotional intelligence<br>• Time management<br>• Lifelong learning |
| William Hargett [32] | Developing a model for effective leadership in healthcare: a concept mapping approach | 2017 | – | Mixed | • Personal integrity<br>• Teambuilding<br>• Professionalism<br>• Fostering vision<br>• Critical thinking<br>• Emotional intelligence<br>• Facilitating transformation |
| Nguyen Duc Thanh [12] | A framework of leadership and managerial competency for preventive health managers in Vietnam | 2021 | Vietnam | Mixed | • Policy development and implementation<br>• Strategy development and implementation<br>• Planning<br>• Risk and disaster management<br>• Quality management<br>• Inspection<br>• Supervision<br>• Monitoring and evaluation<br>• Development of preventive Institutes' vision<br>• Promotion and development of organizational image and public relations<br>• Networking and liaison with stakeholders |
| Kate Wright [33] | Competency development in public health leadership | 2000 | Ameri-can | Review | • Visionary leadership<br>• Sense of mission<br>• Effective change agent<br>• Negotiation<br>• Political awareness<br>• Ethics and power<br>• Marketing and education<br>• Understanding of organizational dynamics<br>• Interorganizational collaboration<br>• Facilitate networking<br>• Social forecasting<br>• Team-building<br>• Facilitate empowerment and motivation |
| Marie Aimee Muhimpundu [10] | Road map for leadership and management in public health: a case study on noncommunicable diseases program managers' training in Rwanda | 2018 | Rwanda | Mixed | Functional Competency:<br>• Mobilize community<br>• Develop policy<br>• Evaluate<br>• Research<br>• Diagnose and investigate<br>Implementation Competency:<br>• Innovation<br>• Partnership<br>• Communication<br>• Political commitment |

*(Continued)*

**Table 3.** (Continued)

| Authors | Title of studies | Year | Country | Design | Competencies |
|---|---|---|---|---|---|
| Darja Kragt [34] | Predicting leadership competency development and promotion among high-potential executives: the role of leader identity | 2020 | Australia | Mixed | • Challenging the status quo<br>• Valuing diversity<br>• Promoting employee voice<br>• Creating commitment<br>• Negotiating<br>• Managing stress<br>• Articulating complex ideas<br>• Adapting to change |
| Judith G. Calhoun [35] | Development of a core competency model for the master of public health degree | 2008 | North America | Mixed | • Communication<br>• Professionalism<br>• Planning<br>• Systems Thinking<br>• Accountability<br>• Leading change<br>• Systems thinking<br>• Partnership<br>• Developing vision<br>• leading mission<br>• Team building<br>• Negotiation<br>• Conflict management<br>• Transparency<br>• Integrity |
| Milica Dikic [36] | Alignment of perceived competencies and perceived job tasks among primary care managers | 2020 | Serbia | Descriptive | • Communication<br>• Team-building<br>• Planning and priority-setting<br>• Performance assessment<br>• Problem-solving<br>• Leading change |
| AwadAllah MB [37] | Managerial competencies of primary health care managers | 2021 | Egypt | Descriptive | • Developing vision<br>• Participative decision making<br>• Motivation |
| The International Hospital Federation [38] | Leadership competencies for healthcare services managers | 2015 | American | Report | • Behavioral competency<br>• Engaging culture and environment<br>• Leading Change<br>• Driving Innovation |

of this is connected to the absence of a comprehensive information system about the burden of diseases, a system for gathering and analyzing information, indicators, and procedures for monitoring and assessment. However, the skills and capabilities of primary care managers in using information and, as a result, decision-making and evidence-informed management are also contributing factors. Other studies claim that by reducing errors, evidence-informed decision-making enhances the quality of management decisions and facilitates the optimal use of limited resources [39,40].

Owing to a lack of a clear vision, proper goal setting, setting priorities, and determining the role of other organizations, significant progress has not been made in the implementation of NCD control programs, despite the development and promulgation of national, provincial, and university documents for the control and prevention of NCDs [41]. Health managers require planning competencies to be able to effectively implement programs. Research from various countries, notably Zimbabwe, demonstrates that district health managers lack the managerial skills needed to carry out their given duties [42]. Planning, risk and disaster management skills were determined to be the most crucial competencies, according to a literature review [15,43].

**Table 4. The consensus and stability between Delphi scoring rounds.**

| Competencies | 1st Delphi scoring | | Consensus (>80) | 2nd Delphi scoring | | Consensus (>80) | Stability p ≥ 0.05 |
|---|---|---|---|---|---|---|---|
| | The mean score of necessity in competencies | The mean score of applicability in competencies | | The mean score of necessity in competencies | The mean score of applicability in competencies | | |
| Communication | 9 | 6.5 | 68.7 | 9 | 8 | 83.3 | 0.14 |
| Driving Innovation | 8.5 | 6 | 62.5 | 9 | 8 | 95.8 | 0.5 |
| Evidence informed decision making | 9 | 7 | 77.5 | 9 | 9 | 97.2 | 0.15 |
| Ethic and Professionalism | 8.5 | 7 | 75 | 9 | 8 | 91.6 | 0.6 |
| Leading and Managing Change | 7 | 6.3 | 74.1 | 8 | 8 | 86 | 0.29 |
| Multisectoral Collaboration | 9 | 6 | 68.7 | 9 | 8.5 | 87.4 | 0.09 |
| Management | 5 | 7 | 62.5 | 9 | 8 | 83.3 | 0. 16 |
| Motivation and inspiration | 8 | 6.5 | 75 | 9 | 8 | 87.4 | 0.6 |
| Political Awareness | 8.5 | 6.5 | 71.2 | 8.5 | 8 | 88.3 | 0.11 |
| Planning | 8.5 | 6.2 | 87.5 | 9 | 7 | 96.4 | 0.1 |
| Personality | 8 | 6 | 62.5 | 9 | 8 | 91.6 | 0.29 |
| Quality Improvement | 9 | 7 | 75 | 9 | 8 | 83.3 | 0.07 |
| Risk and Disaster Management | 8 | 7 | 62.5 | 8 | 7 | 85 | 0.24 |
| Systematic Thinking | 8 | 6 | 68.7 | 9 | 8 | 89.1 | 0.2 |
| Team Building | 9 | 7 | 75 | 9 | 8 | 83.3 | 0.3 |

## Distinctive domain

The distinctive competencies of being innovative, leading and managing change, team building, communication, quality improvement and systematic thinking were also identified as important. One of the skills required of leaders in the field of PHC is the ability to identify and involve various stakeholders in a variety of interdisciplinary projects, which is consistent with the management philosophy based on participation to improve public health. It is also important to ensure the continued performance of units under supervision in the face of changes made to the primary care health system [44].

In the Iranian health system, significant variations and disruptions in the regular scheduling required in NCD programs are caused by frequent changes in management positions, particularly at the ministry level, a lack of trained and skilled workers, and the promulgation of multiple overlapping but separate programs. An additional challenge has been caused by the various political and economic sanctions that have impacted the supply and distribution of some medications and equipment [16,45]. Under such circumstances, there is an even greater need for managers with expertise and skills as change agents to adapt programs appropriately [16]. Improving health literacy and increasing societal expectations of quality and safety in health care programs are other drivers of the need for empowered and well-trained managers. This trend is recognized globally [15,46]. Inadequate planning input from primary healthcare professionals has been demonstrated to lower the effectiveness and success rate of public health programs [47]. Consequently, achieving the best outcomes of NCD programs

**Table 5. List of leadership domains and competencies.**

| Domains | Competency | Detailed description |
|---|---|---|
| Behavioral | Ethic and Professionalism | - Having the knowledge, ability and attitude to apply the principles of professional ethical values in different situations, expressing effective conflicts of interest in all executive stages and making decisions for the control and prevention programs of NCDs |
| | | - Having the knowledge, ability and attitude in creating a vision, commitment to set goals and values, and professional responsibility in line with the implementation of the control and prevention programs of NCDs |
| | Motivation and Inspiration | - Having the knowledge, ability and attitude to inspire oneself and the subordinate units to achieve the vision and goals set in the control and prevention programs of NCDs |
| | | - Having the knowledge, ability and attitude in the rational use of reward and punishment mechanisms, openness to new opinions and perspectives and attention to sensitivities and cultural differences in order to stimulate innovation in the implementation of the control and prevention programs of NCDs |
| | Personality | Having personality traits such as transparency, trust, fairness, independence, power of influence, truthfulness and authority |
| Distinct | Driving Innovation | - Having the knowledge, ability and attitude to use creative methods and solutions in solving problems and making decisions related the control and prevention programs of NCDs |
| | | - Having the knowledge, ability and attitude in stimulating creativity and innovation, accepting and managing diverse opinions and suggestions of all stakeholders involved in the implementation of the control and prevention programs of NCDs |
| | Leading and Managing Change | - Having the knowledge, ability and attitude to identify opportunities for changes in the structure of the subunits, trends and processes related to the implementation of the control and prevention programs of NCDs |
| | | - Having the knowledge, ability, and attitude in reviewing the mission and setting goals related to the control and prevention programs of NCDs in times of political, social, and economic changes. |
| | | - Having the knowledge, ability and attitude to effectively guide employees and resources related to the control and prevention programs of NCDs and ensure the continuity of the team and subunits in dealing with the changes (sanctions, political, economic, social, crises) |
| | Team Building | - Having the knowledge, ability and attitude to work with others and change process structures and data in order to form and put together effective and multidisciplinary work teams to implement measures related to the control and prevention programs of NCDs |
| | | - Having the knowledge, ability and attitude in clarifying and adjusting roles and duties, facilitating the provision of facilities and resources, facilitating communication processes between working groups and sharing experiences and successes, promoting continuous learning of the formed teams related to the control and prevention programs of NCDs |
| | Communication | - Having the knowledge, ability and attitude to maintain positive and continuous relations with influential stakeholders (community representatives, people, local leaders, public and private bodies and organizations) in the implementation of the control and prevention programs of NCDs |
| | | - Having the knowledge, ability and attitude to negotiate and bargain with different stakeholders (especially political) in the implementation of the control and prevention programs of NCDs |
| | Quality Improvement | - Having knowledge, ability and attitude in regular monitoring and identification of bottlenecks related to various processes of service delivery, noncompliance with the established standards and guidelines, applying quality improvement methods and promoting continuous learning related to the implementation of the control and prevention programs of NCDs |
| | Systematic Thinking | - Having the knowledge, ability and attitude of analytical and conceptual thinking in evaluating and solving problems related to the control and prevention programs of NCDs |
| | | - Having the knowledge, ability and attitude to combine and integrate the views of various influential stakeholders, other influential areas in the implementation of the control and prevention programs of NCDs |
| Managerial | Management of resources, executive, and administrative functions | Having the knowledge, ability and attitude in performing managerial tasks including financial affairs and distribution of financial resources, human, material and physical resources, information and administrative correspondence in the job position of manager of NCDs unit |
| Specific | Multisectoral Collaboration | - Having the knowledge, ability and attitude to mobilize and maintain the participation of governmental and nongovernmental organizations (private, quasi-governmental) in order to identify and solve problems related to the field of NCDs |
| | | - Having the knowledge, ability and attitude to identify, evaluate and analyze areas that require multiple interventions by governmental and nongovernmental organizations (private, quasi-governmental) in NCDs control and prevention programs |

*(Continued)*

**Table 5.** (Continued)

| Domains | Competency | Detailed description |
|---|---|---|
| | Political Awareness | - Having the knowledge, ability and attitude to identify and apply appropriate measures in the implementation of NCDs control and prevention programs, taking political considerations into account |
| | | - Having the knowledge, ability and attitude to create political commitment to attract resources and support in the implementation of interventions related to the control and prevention of NCDs |
| | Evidence informed decision making | - Having the knowledge, ability and attitude to use the information obtained from the analysis of indicators and vital statistics in determining the goals, consequences and how to provide services related to the control and prevention programs of NCDs |
| | | - Having the knowledge, ability and attitude in analyzing the current situation (for example, in terms of epidemiology of diseases), determining needs and priorities in accordance with the information obtained from the evaluation of indicators and the care system of NCDs, and finally providing objective and based solutions on evidence to solve problems related to the control and prevention programs of NCDs |
| | | - Having the knowledge, ability and attitude toward laws and regulations, policies and upstream documents related to the health system and especially the field of NCDs |
| | Risk and Disaster Management | - Having the knowledge, ability and attitude in the field of crisis and emergency planning, determining measures and interventions, resources to maintain the continuity of service provision and implementing programs related to the control and prevention programs of NCDs |
| | Planning | - Having the knowledge, ability and attitude to determine general and specific goals using the results of monitoring the NCDs care system, indicators and other vital statistics related to the control and prevention programs of NCDs |
| | | - Having the knowledge, ability and attitude to determine appropriate indicators to monitor interventions and activities in order to achieve the set goals related to the control and prevention programs of NCDs |
| | | - Having the knowledge, ability and attitude in formulating a strategic and operational plan in line with the implementation of interventions related to control and prevention programs of NCDs |
| | | - Having the knowledge, ability and attitude in analyzing the strategic position of the province and region, analyzing different stakeholders related to the control and prevention programs of NCDs |
| | | - Having the knowledge, ability and attitude to estimate the resources (both human, financial and equipment) necessary in the implementation of interventions related to the control and prevention programs of NCDs |
| | | - Having the knowledge, ability and attitude in monitoring and evaluating the goals and indicators set in the control and prevention programs of NCDs |
| | | - Having the knowledge, ability and attitude in regular and structured feedback of the expected results and the achievements determined in the control and prevention programs of NCDs |

with available resources requires managers to be accepting of and skilled in teamwork and the ability to use collective wisdom [10].

## Managerial domain

The manager of the NCD unit requires managerial competencies that include knowledge, ability, and attitudes when performing managerial tasks. Studies conducted in Kenya and Zimbabwe indicate that almost all public health managers reported having insufficient management skills to perform their management duties effectively. They may not perform their duties properly if they lack adequate management training [42]. Technical competence is necessary for health managers to perform the tasks expected of them. Skills in supervision, financial administration, and the distribution of financial resources, human resources, material resources, and physical resources are necessary for health management competency [42,48].

## Behavioral domain

The competencies and descriptive statements for the competencies of ethics and professionalism, motivation and inspiration, and personality are covered in the behavioral domain. Other research at PHC organizations in Iran has demonstrated that staff underperformance

and job burnout are the results of unequal payments [39]. Additionally, the lack of a clear job description and career path, as well as the notification of numerous programs from higher levels without considering the capacity of human resources, particularly in programs for the prevention and control of NCDs, have led to dissatisfaction and problems with motivation among the program's staff members and managers. Consequently, managers need behavioral competencies to address these types of issues and increase employee satisfaction and motivation [25].

The strengths of this research are the use of a comprehensive methodology through a literature review, the purposive selection of experts nationwide, and the high scores obtained for leadership competencies in the Delphi technique phase. Another strength of this research was the identification of specific leadership competencies for managers of prevention and control programs for NCDs in Iran for the first time. The fifteen leadership competencies can provide an effective tool for assessing the performance of program managers in the prevention and control of NCDs. They can also be utilized as training courses to empower program managers of NCDs at the national level. The study's limitations were due to the type of participants. Noncommunicable disease managers' opinions at other levels, such as those of regional managers or former managers, were not included in the study. Another limitation of this research was the cooperation of the participants, which was solved by relying on organizational correspondence, continuous follow-up, and justifying the importance of the research. The final limitation of this research was the lack of prioritizing leadership competencies through prioritizing methods.

## Conclusion

The present study shows that managers of prevention and control programs for NCDs should be equipped with leadership competencies such as multisectoral collaboration, political awareness, evidence-informed decision making, risk and disaster management, planning, driving innovation, leading and managing change, team building, communication, quality improvement, systematic thinking, management, ethics and professionalism, motivation and inspiration and personality. These fifteen leadership competencies can be utilized as helpful tools and guides for assessing managers and developing educational resources for training programs to improve managers' leadership.

## Supporting information

**S1 Checklist.   COREQ checklist.**
(DOCX)

## Acknowledgments

The authors are grateful to all the participants who contributed to this study.

## Author contributions

**Conceptualization:** Jafar Sadegh Tabrizi, Yegane Partovi, Kamal Gholipour, Mostafa Farahbakhsh, Tohid Jafari Koshki, Ahmad Koosha, Jabreil Sharbafi.

**Formal analysis:** Jafar Sadegh Tabrizi, Yegane Partovi.

**Investigation:** Jafar Sadegh Tabrizi, Yegane Partovi, Andrew Wilson.

**Methodology:** Jafar Sadegh Tabrizi, Yegane Partovi, Andrew Wilson, Kamal Gholipour.

**Project administration:** Yegane Partovi.

**Resources:** Yegane Partovi.

**Supervision:** Jafar Sadegh Tabrizi, Yegane Partovi.

**Validation:** Jafar Sadegh Tabrizi, Yegane Partovi, Mostafa Farahbakhsh, Tohid Jafari Koshki.

**Writing – original draft:** Jafar Sadegh Tabrizi, Yegane Partovi.

**Writing – review & editing:** Jafar Sadegh Tabrizi, Yegane Partovi, Andrew Wilson, Kamal Gholipour, Tohid Jafari Koshki, Ahmad Koosha, Jabreil Sharbafi.

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
