## [Decision Letter · Decision Letter 0]

12 Dec 2023

PONE-D-23-29501Developing Core Leadership Competencies to Success Non-Communicable Disease Control and Prevention Programs: A Mixed- Methods StudyPLOS ONE

Dear Dr. partovi,

Thank you for submitting your manuscript to PLOS ONE. After careful consideration, we feel that it has merit but does not fully meet PLOS ONE’s publication criteria as it currently stands. Therefore, we invite you to submit a revised version of the manuscript that addresses the points raised during the review process.

We look forward to receiving your revised manuscript.

Kind regards,

Edris Hasanpoor

Academic Editor

PLOS ONE

Journal Requirements:

Reviewers' comments:

Reviewer's Responses to Questions

**Comments to the Author**

1. Is the manuscript technically sound, and do the data support the conclusions?

Reviewer #1: Yes

Reviewer #2: Yes

2. Has the statistical analysis been performed appropriately and rigorously? 

Reviewer #1: I Don't Know

Reviewer #2: No

3. Have the authors made all data underlying the findings in their manuscript fully available?

Reviewer #1: No

Reviewer #2: No

4. Is the manuscript presented in an intelligible fashion and written in standard English?

Reviewer #1: Yes

Reviewer #2: Yes

5. Review Comments to the Author

Reviewer #1: Abstract

1. It seems that in this study, "Core Leadership Competencies to Success Non-Communicable Disease Control and Prevention Programs", have only been identified; And the word "...Developing" is beyond and wider than what is done in this study. If the authors' team deems it appropriate, I suggest that a more appropriate word be substituted for the term "Develop*" in the entire article (title, abstract and main text).

2. In the results section of the abstract, "inspiration" is written as one of the main competencies, while it is not mentioned in Table 2, which lists the competencies. Please be consistent about it throughout the article.

3. Please use a specific term, and not its synonyms, in introducing and talking about leadership competencies throughout the article. For example, although you have used the term "planning" in the abstract, you have used the term "Plan Making" in other parts of the article. Please maintain consistency.

4. In writing the purpose of the article (This study developed leadership competencies for managers of non-communicable diseases in Iran.), I suggest that it be clearly stated that the purpose of the study is to identify these competencies.

***

Introduction

1. Please correct the following sentence by writing more up-to-date statistics: "According to statistics from 2019, NCDs were the cause of 74.4% of all deaths and 63.8% of years lived with a disability (1).”

2. In connecting the two main issues of the study (i.e., "non-communicable diseases" and "leadership competencies"), there is a lot of content weakness. The authors have used the following phrase to connect these two issues, while it has no reference and does not seem very appropriate.

(These capacities are essential prerequisites to implement leadership skills who can coordinate, support, and ensure their long-term continuity in order to produce beneficial outcomes. Leaders with required competencies and qualifications is important as essential to ensure effective interventions are delivered as efficiently as possible within the available resources, especially in low- and middle-income countries.)

Please use specific scientific references in creating a connection between these two main issues, and connect them with a literature justifying them.

3. Please rewrite and correct the following parts by referring to scientific references.

(In order to develop leadership and management expertise in those who will be responsible for the implementation and running of NCD prevention and control programs, it is essential to identify the required competencies. Globally, existing educational programs for primary health care (PHC) managers lack clear leadership and management competencies for PCNCD program5 management.)

4. Please provide evidence of the failure of specific health programs in Iran and the world, that the cause of the failure was actually ineffective management.

***

Methods

1. Inclusion and exclusion criteria related to literature review should be written.

***

Results

1. In writing the title of Table 3, the word "remaining" is not necessary.

2. The explanations related to Table 3 should be written above it.

***

Conclusion

1. The following text needs a reference. (Planning, risk, and disaster management skills were determined to be the most crucial competencies, according to the literature review.)

2. Please remove the sentences without reference (except in permitted cases) from the text of the manuscript or rewrite them based on a valid reference (for example: Consequently, to achieve the best outcomes of NCDs programs with available resources requires managers accepting of and skilled in teamwork and the ability for using collective wisdom.)

3. In general, the discussion section does not meet my expectations of a good discussion. Please discuss each of the four areas ('specific', 'distinct', 'managerial' and 'behavioral') in 4 separate paragraphs, write about the leadership competencies in each area and their supporting evidence. Write more clearly about the area under discussion and the competencies covered by it.

4. Strengths and limitations of the study are not mentioned.

Reviewer #2: Dear Authors,

I have concluded my review with minor revisions to share. The introduction offers a fond overview to the relevance of NCDs and their health-economic implications. I have no further remarks to the motivation and discussion.

Following remarks should be considered for the methodology:

- Was an appropriate study guideline followed? If so please indicate in the text or carry out this step. One of the possibilities should be the SQUIRE Guideline

- Ll.149 According to the PLOSONE Guidelines the identified studies should be shared in order to provide a transparent methodology. Also please report the search strategy and consider the use of a flow diagram and explain the concatenation of the search strings (see below)

- Ll.149 Please explain the revision of competencies more precisely, respectively why there was no change in the absolute numbers

- L.173 The source on the DELPHI panel procedure is unclear. Please cite in the text wat (modified) methodology was used and what the requirements are.

- L.177 sentence structure not straight forward

- L.195 Please explain precisely how many rounds were made if not two, respectively how the final decision was made

- L.217 Where does the “19” come from?

- L. 241 Having introduced the first set of competencies is a bold assumption, especially where the overall topic is not restricted to Iran. Please state explicitly what new findings the study brings.

- Table 1: What makes the demographic distribution a relevant result. Use tables for the more relevant results such as the identified competencies, a flow diagram of the research and how many competencies were included in your sources (average and histological distribution, adding to the point made about line 241)

- Table 2: Give the competencies ranked by importance.

- Table 3: Adjust the formatting uniformly, especially the use of “-“

Best regards.

6. PLOS authors have the option to publish the peer review history of their article (what does this mean? ). If published, this will include your full peer review and any attached files.

**Do you want your identity to be public for this peer review?** For information about this choice, including consent withdrawal, please see our Privacy Policy .

Reviewer #1: No

Reviewer #2: No

---

## [Author Response · Author response to Decision Letter 1]

7 Jun 2024

Point-by-point response to reviewers

Dear Editor- in- Chief

Our thanks to the reviewers and the subject editor for thoughtful critiques of our manuscript. We have adopted all of the suggestions. We think that the manuscript has been greatly improved by these revisions and we hope that you will now find it suitable for publication in PLOSONE. Our point-by-point responses to comments are detailed on the following pages.

Best regards,

Yegane Partovi

Corresponding Author

Response to academic editor:

In your Data Availability statement, you have not specified where the minimal data set underlying the results described in your manuscript can be found. PLOS defines a study's minimal data set as the underlying data used to reach the conclusions drawn in the manuscript and any additional data required to replicate the reported study findings in their entirety. All PLOS journals require that the minimal data set be made fully available. For more information about our data policy, please see http://journals.plos.org/plosone/s/data-availability. All of attachment was provided in supplementary file1.

Please include your full ethics statement in the ‘Methods’ section of your manuscript file. In your statement, please include the full name of the IRB or ethics committee who approved or waived your study, as well as whether or not you obtained informed written or verbal consent. If consent was waived for your study, please include this information in your statement as well. full ethics statement in the ‘Methods’ section was added.

Reviewer #1: Abstract

1. It seems that in this study, "Core Leadership Competencies to Success Non-Communicable Disease Control and Prevention Programs", have only been identified; And the word "...Developing" is beyond and wider than what is done in this study. If the authors' team deems it appropriate, I suggest that a more appropriate word be substituted for the term "Develop*" in the entire article (title, abstract and main text). The word 'identify' has been substituted for the term 'develop' in the entire article (title, abstract and main text). Amendments are indicated in the text by yellow highlighting.

2. In the results section of the abstract, "inspiration" is written as one of the main competencies, while it is not mentioned in Table 2, which lists the competencies. Please be consistent about it throughout the article. Amendments have been made. They are indicated in the text by yellow highlighting.

3. Please use a specific term, and not its synonyms, in introducing and talking about leadership competencies throughout the article. For example, although you have used the term "planning" in the abstract, you have used the term "Plan Making" in other parts of the article. Please maintain consistency. Amendments have been made. They are indicated in the text by yellow highlighting.

4. In writing the purpose of the article (This study developed leadership competencies for managers of non-communicable diseases in Iran.), I suggest that it be clearly stated that the purpose of the study is to identify these competencies. Amendments have been made. They are indicated in the text by yellow highlighting.

Introduction

1. Please correct the following sentence by writing more up-to-date statistics: "According to statistics from 2019, NCDs were the cause of 74.4% of all deaths and 63.8% of years lived with a disability (1).” The latest statistics provided in 2019 are cited in the article text according to the research conducted by this study's researchers. All the reliable and relevant sources in this field including WHO, WORLD BANK….. were examined. The latest statistics are for 2019.

2. In connecting the two main issues of the study (i.e., "non-communicable diseases" and "leadership competencies"), there is a lot of content weakness. The authors have used the following phrase to connect these two issues, while it has no reference and does not seem very appropriate.

(These capacities are essential prerequisites to implement leadership skills who can coordinate, support, and ensure their long-term continuity in order to produce beneficial outcomes. Leaders with required competencies and qualifications is important as essential to ensure effective interventions are delivered as efficiently as possible within the available resources, especially in low- and middle-income countries.)

Please use specific scientific references in creating a connection between these two main issues, and connect them with a literature justifying them. Amendments have been made. They are indicated in the text by yellow highlighting.

3. Please rewrite and correct the following parts by referring to scientific references.

(In order to develop leadership and management expertise in those who will be responsible for the implementation and running of NCD prevention and control programs, it is essential to identify the required competencies. Globally, existing educational programs for primary health care (PHC) managers lack clear leadership and management competencies for PCNCD program5 management.) Amendments have been made. They are indicated in the text by yellow highlighting.

4. Please provide evidence of the failure of specific health programs in Iran and the world, that the cause of the failure was actually ineffective management. The main purpose of this study is not to investigate managerial inefficiency and the failure of health programs due to the inefficiency of managers. The current education programs for PHC managers worldwide do not provide clear leadership competencies for NCD program management.

***

Methods

1. Inclusion and exclusion criteria related to literature review should be written. Amendments have been made. They are indicated in the text by yellow highlighting.

***

Results

1. In writing the title of Table 3, the word "remaining" is not necessary. Amendments have been made.

2. The explanations related to Table 3 should be written above it. Amendments have been made.

They are indicated in the text by yellow highlighting.

***

Conclusion

1. The following text needs a reference. (Planning, risk, and disaster management skills were determined to be the most crucial competencies, according to the literature review.) Some references have been added. They are indicated in the text by yellow highlighting.

2. Please remove the sentences without reference (except in permitted cases) from the text of the manuscript or rewrite them based on a valid reference (for example: Consequently, to achieve the best outcomes of NCDs programs with available resources requires managers accepting of and skilled in teamwork and the ability for using collective wisdom.) Some references have been added. They are indicated in the text by yellow highlighting.

3. In general, the discussion section does not meet my expectations of a good discussion. Please discuss each of the four areas ('specific', 'distinct', 'managerial' and 'behavioral') in 4 separate paragraphs, write about the leadership competencies in each area and their supporting evidence. Write more clearly about the area under discussion and the competencies covered by it.

4. Strengths and limitations of the study are not mentioned. Amendments have been made.

They are indicated in the text by yellow highlighting.

Reviewer #2: Dear Authors,

I have concluded my review with minor revisions to share. The introduction offers a fond overview to the relevance of NCDs and their health-economic implications. I have no further remarks to the motivation and discussion.

Following remarks should be considered for the methodology:

- Was an appropriate study guideline followed? If so, please indicate in the text or carry out this step. One of the possibilities should be the SQUIRE Guideline. Amendments have been made.

They are indicated in the text by green highlighting.

- Ll.149 According to the PLOSONE Guidelines the identified studies should be shared in order to provide a transparent methodology. Also please report the search strategy and consider the use of a flow diagram and explain the concatenation of the search strings (see below) Amendments have been made. The identified studies provided in Attachment file 1.

- Ll.149 Please explain the revision of competencies more precisely, respectively why there was no change in the absolute numbers. Amendments have been made. They provided in table1.

- L.173 The source on the DELPHI panel procedure is unclear. Please cite in the text wat (modified) methodology was used. Some references have been added. They are indicated in the text by green highlighting.

- L.177 sentence structure not straight forward. . Amendments have been made.

- L.195 Please explain precisely how many rounds were made if not two, respectively how the final decision was made. Amendments have been made. They are indicated in the text by green highlighting.

The 19 leadership competencies entered the first round of the Delphi phase with near 90% of invited participants responding. The titles of 4 domains were agreed upon by all participants. The ‘risk management’ competency was merged with the ‘disaster management’ competency and the competencies of ‘resource, executive and administrative management’ were merged as ‘planning’. The ‘managing change’ competency was rephrased into ‘leading and managing change’. Competencies that received 80 to 100% agreement in the first round of the Delphi process were accepted without further testing, while those that received lower than 80% agreement underwent revision based on participant comments. Four domains and fifteen leadership competencies were presented for the second round of Delphi after modification and editing. Again, there was no disagreement on the four fields. We concluded that saturation and consensus had been reached because no additional competencies were proposed in the second round.

- L.217 Where does the “19” come from? Amendments have been made. They provided in table1.

- L. 241 Having introduced the first set of competencies is a bold assumption, especially where the overall topic is not restricted to Iran. Please state explicitly what new findings the study brings.

Globally, a few educational programs have been created and put into place to improve the leadership and management competencies of primary health care managers, but none of them have specifically addressed the leadership competencies needed for managers of NCDs. As a result, our study developed leadership competencies for managers of non-communicable diseases in Iran for the first time.

- Table 3: Give the competencies ranked by importance. The objective of this study was to identify the leadership competences required for managers of non-communicable diseases. Prioritizing the competencies identified in the present study using the AHP method is possible, but it wasn't one of our objectives.

- Table 4: Adjust the formatting uniformly, especially the use of “-“ Amendments have been made.

---

## [Decision Letter · Decision Letter 1]

6 Aug 2024

PONE-D-23-29501R1Identifying Core Leadership Competencies to Success Non-Communicable Disease Control and Prevention Programs: A Mixed- Methods StudyPLOS ONE

Dear Dr. partovi,

Thank you for submitting your manuscript to PLOS ONE. After careful consideration, we feel that it has merit but does not fully meet PLOS ONE’s publication criteria as it currently stands. Therefore, we invite you to submit a revised version of the manuscript that addresses the points raised during the review process.

Please see the comments below of two reviewers and respond to their feedback in your resubmission. 

We look forward to receiving your revised manuscript.

Kind regards,

Joanna Tindall

Staff Editor

PLOS ONE

Journal Requirements:

Reviewers' comments:

Reviewer's Responses to Questions

**Comments to the Author**

1. If the authors have adequately addressed your comments raised in a previous round of review and you feel that this manuscript is now acceptable for publication, you may indicate that here to bypass the “Comments to the Author” section, enter your conflict of interest statement in the “Confidential to Editor” section, and submit your "Accept" recommendation.

Reviewer #1: All comments have been addressed

Reviewer #2: (No Response)

2. Is the manuscript technically sound, and do the data support the conclusions?

Reviewer #1: Yes

Reviewer #2: Yes

3. Has the statistical analysis been performed appropriately and rigorously? 

Reviewer #1: I Don't Know

Reviewer #2: I Don't Know

4. Have the authors made all data underlying the findings in their manuscript fully available?

Reviewer #1: Yes

Reviewer #2: No

5. Is the manuscript presented in an intelligible fashion and written in standard English?

Reviewer #1: Yes

Reviewer #2: No

6. Review Comments to the Author

Reviewer #1: 1. In lines 70 and 72, in two consecutive sentences, the word ""According to"" is repeated; Please avoid repetition.

2. Although you have added a sentence about the strengths and weaknesses of the research, it needs further elaboration. Please write more about the strengths and weaknesses of the research.

Reviewer #2: Overall statement: Many of my previous comments were adressed, though not all. I have marked some additional problems in language, but certtainly a precise reread of the authors is necessary to eliminate formatting and language errors, otherwise it does not seem like the critics were taken seriously. Moreover the methodological description lacks the necessary clarity, see comments below. I strongly recommend the use of clearer flowcharts for 1. Literature process and 2. Competencies process.

Abstract: No comments

Introduction:

Please use primary sources when citing counts and costs of NCD. In particular the way reference (2) is used is not valid.

The statements made about NCD should be more conditional unless proven upon no doubt.

Methods:

The filled in COREQ checklist should be provided in appendix.

Again consider the use of a flow diagram. How many studies were duplicates, how many studies were screened? Were the keywords used separately or with OR/AND concatenation?

What were the main results of exclusion? Table 1 is more a figure that should be focussing more on visualizing the flow from 150 to 15. Duplicates merge etc. should be reported clearly

State the reason for dividing expert and Delphi round and the main difference. Was consensus in the DELPHI panel assessed? Why?

Was an 80% response rate reached? Were do you explain the consensus from table 1? Ll.199 not clear

Results:

Consider including all 19 competencies in table 3 for clarity. How is table 3 ordered? Consider stating by domains and alphabetically

Are the included papers cited in the appendix?

Language:

l.71 insert space before (1)

ll.82 Make clear (with source) that the UN MDG is in danger, the threatening spread is a logical consequence

l. 80 insert space

l. 93 formatting of citation

l. 112 formatting

l. 113 as far as known to the authors?

l. 118 remove space, insert space

l. 120 between … and

l. 135 of the Tabriz?

l. 136 remove space

l. 147 or English

l. 179 10 experts according to the methodology you followed; 20 of the 27 participants?

l. 184 language

l. 193 language

l. 234 Management � Resource, executive and administrative management?

7. PLOS authors have the option to publish the peer review history of their article (what does this mean? ). If published, this will include your full peer review and any attached files.

**Do you want your identity to be public for this peer review?** For information about this choice, including consent withdrawal, please see our Privacy Policy .

Reviewer #1: No

Reviewer #2: No

---

## [Author Response · Author response to Decision Letter 2]

22 Aug 2024

Point-by-point response to reviewers

Dear Editor- in- Chief

Our thanks to the reviewers and the subject editor for thoughtful critiques of our manuscript. We have adopted all of the suggestions. We think that the manuscript has been greatly improved by these revisions and we hope that you will now find it suitable for publication in PLOSONE. Our point-by-point responses to comments are detailed on the following pages.

Best regards,

Yegane Partovi

Corresponding Author

Reviewer #1:

We appreciate the valuable feedback from you, which has significantly improved our work.

1. In lines 70 and 72, in two consecutive sentences, the word ""According to"" is repeated; Please avoid repetition. Amendments have been made. They are indicated in the text by gray highlighting.

2. Although you have added a sentence about the strengths and weaknesses of the research, it needs further elaboration. Please write more about the strengths and weaknesses of the research. The strengths and weaknesses of the research have been added. They are indicated in the text by gray highlighting.

Reviewer #2:

Thank you for your insightful comment. We appreciate the valuable feedback from you, which has significantly improved our work. We try to provide the valid response based on our knowledge. We would be grateful if you could kindly accept our responses.

Overall statement: Many of my previous comments were addressed, though not all. I have marked some additional problems in language, but certainly a precise reread of the authors is necessary to eliminate formatting and language errors, otherwise it does not seem like the critics were taken seriously. Moreover, the methodological description lacks the necessary clarity, see comments below. I strongly recommend the use of clearer flowcharts for 1. Literature process and 2. Competencies process. Amendments have been made. Fig 1 and fig 2 have been added. They are indicated in the text by purple highlighting.

Introduction:

Please use primary sources when citing counts and costs of NCD. In particular, the way reference (2) is used is not valid. Amendments have been made. They are indicated in the text by purple highlighting.

The statements made about NCD should be more conditional unless proven upon no doubt.

Methods:

The filled in COREQ checklist should be provided in appendix. It was provided.

Again consider the use of a flow diagram. How many studies were duplicates, how many studies were screened? Were the keywords used separately or with OR/AND concatenation? What were the main results of exclusion? Amendments have been made. Fig 1 and table 1 (the search strategy) have been added in method section. They are indicated in the text by purple highlighting.

Table 1 is more a figure that should be focussing more on visualizing the flow from 150 to 15. Duplicates merge etc. should be reported clearly. Fig 2 has been included. They are indicated in the text by purple highlighting.

State the reason for dividing expert and Delphi round and the main difference. Was consensus in the DELPHI panel assessed? Why? Two in-person expert panels (for 60 minutes each time) were held to evaluate, screening and make changes to the initial draft leadership competencies. consensus was assessed in the DELPHI panel. Amendments have been made. You can see them in expert panel and Delphi technique survey sections. They are indicated in the text by purple highlighting.

Was an 80% response rate reached? The consensus was reached when 80% of the participants rated a competency above 7 in each round. The response rate of the participants near 90%. Were do you explain the consensus from table 1? we explain it in Delphi technique survey sections. Competencies that received 80 to 100% agreement in the first round of the Delphi process were accepted without further testing, while those that received lower than 80% agreement underwent revision based on participant comments. They are indicated in the text by purple highlighting.

Results:

Consider including all 15 competencies in table 3 for clarity. How is table 3 ordered? Consider stating by domains and alphabetically. We appreciate this suggestion. Table 4 and table 5 have been modified by alphabetically. They are indicated in the text by purple highlighting.

Are the included papers cited in the appendix? 15 references have been added.

l.71 insert space before (1). They are indicated in the text by purple highlighting.

ll.82 Make clear (with source) that the UN MDG is in danger, the threatening spread is a logical consequence. They are indicated in the text by purple highlighting.

l. 80 insert space. They are indicated in the text by purple highlighting.

l. 93 formatting of citation. They are indicated in the text by purple highlighting.

l. 112 formatting. They are indicated in the text by purple highlighting.

l. 113 as far as known to the authors? They are indicated in the text by purple highlighting.

l. 118 remove space, insert space They are indicated in the text by purple highlighting.

l. 120 between … and They are indicated in the text by purple highlighting.

l. 135 of the Tabriz? They are indicated in the text by purple highlighting.

l. 136 remove space They are indicated in the text by purple highlighting.

l. 147 or English They are indicated in the text by purple highlighting.

l. 184 language They are indicated in the text by purple highlighting.

l. 193 language They are indicated in the text by purple highlighting.

l. 234 Management � Resource, executive and administrative management? They are indicated in the text by purple highlighting.

---

## [Decision Letter · Decision Letter 2]

1 Oct 2024

PONE-D-23-29501R2Identifying Core Leadership Competencies to Success Non-Communicable Disease Control and Prevention Programs: A Mixed- Methods StudyPLOS ONE

Dear Dr. partovi,

Thank you for submitting your manuscript to PLOS ONE. After careful consideration, we feel that it has merit but does not fully meet PLOS ONE’s publication criteria as it currently stands. Therefore, we invite you to submit a revised version of the manuscript that addresses the points raised during the review process.

We look forward to receiving your revised manuscript.

Kind regards,

Dorothy Lall

Academic Editor

PLOS ONE

Journal Requirements:

Reviewers' comments:

Reviewer's Responses to Questions

**Comments to the Author**

1. If the authors have adequately addressed your comments raised in a previous round of review and you feel that this manuscript is now acceptable for publication, you may indicate that here to bypass the “Comments to the Author” section, enter your conflict of interest statement in the “Confidential to Editor” section, and submit your "Accept" recommendation.

Reviewer #1: All comments have been addressed

Reviewer #2: (No Response)

2. Is the manuscript technically sound, and do the data support the conclusions?

Reviewer #1: Yes

Reviewer #2: Yes

3. Has the statistical analysis been performed appropriately and rigorously? 

Reviewer #1: I Don't Know

Reviewer #2: I Don't Know

4. Have the authors made all data underlying the findings in their manuscript fully available?

Reviewer #1: No

Reviewer #2: Yes

5. Is the manuscript presented in an intelligible fashion and written in standard English?

Reviewer #1: Yes

Reviewer #2: No

6. Review Comments to the Author

Reviewer #1: Recommended corrections are well done, all corrections have been addressed. I thank the authors for making these corrections.

Reviewer #2: Dear Authors, dear Editor,

the comments I have made in the frist revision have been largely adressed. However there are still points that I have to note, some of them for the third time:

1. Language. I have made some suggestions previously, but suggested proofread by the authors. It appears that there was no effort to go beyond the minimum reuirements of my comments. Errors persist such as competences/competencies; NCD not introduced in introduction, new limitation section overall not fluent; Table 1 minning spaces, title mispelled, cursive; Figure 1 in bad quality (perhaps due to upload); reports excluded not in uniform fashion, use of "=", " =", " = "; Appendix named Delphii. In the way the manuscript is presented I do not feel that the language concerns are taken serious in any way.

2. Methodology 150 to 19. You state that

a) Based on the expert panel input, the competencies were categorizing into four domains titled specific, distinct, managerial and behavioral.

b) Based on the Delphi results, the competencies were categorized into four domains titled specific, distinct, managerial and behavioral

Apart from the error of "categorizing", it is yet to be explaied where the 19 domains come from (based on the 150) or where the difference of these sentences lies. The grouping of domains and categorizing are (in my understanding) two different steps. Figure 2 is unfortunately of no help in understanding the process. I recommend to delete the figure and replace it by a) a better flow of competencies or b) extended explanation in text.

In the current way of responding to my reviews this unfortunately leaves me with more revisions to make.

Best.

7. PLOS authors have the option to publish the peer review history of their article (what does this mean? ). If published, this will include your full peer review and any attached files.

**Do you want your identity to be public for this peer review?** For information about this choice, including consent withdrawal, please see our Privacy Policy .

Reviewer #1: No

Reviewer #2: No

---

## [Author Response · Author response to Decision Letter 3]

28 Dec 2024

“Reviewer #1: Recommended corrections are well done, all corrections have been addressed. I thank the authors for making these corrections. Thank you.

Reviewer #2: Dear Authors, dear Editor,

the comments I have made in the frist revision have been largely adressed. However there are still points that I have to note, some of them for the third time:

1. Language. I have made some suggestions previously, but suggested proofread by the authors. It appears that there was no effort to go beyond the minimum reuirements of my comments. Errors persist such as competences/competencies; NCD not introduced in introduction, new limitation section overall not fluent; Table 1 mining spaces, title misspelled, cursive; Figure 1 in bad quality (perhaps due to upload); reports excluded not in uniform fashion, use of "=", " =", " = "; Appendix named Delphii. In the way the manuscript is presented I do not feel that the language concerns are taken serious in any way.

I hope this message finds you well. As we welcome the New Year, I wanted to take a moment to express my heartfelt gratitude for your valuable contributions as a reviewer. Your insights and feedback have been instrumental in enhancing the quality of our work. Wishing you a joyful and prosperous New Year filled with success and happiness. May 2025 bring you new opportunities and continued achievements.

The Prisma is written according to the provided Prisma 2020 template (https://www.prisma-statement.org/prisma-2020-flow-diagram).

The text of the manuscript was reviewed by a linguistic editor and changes were made. The changes are highlighted in turquoise color.

2. Methodology 150 to 19. You state that

a) Based on the expert panel input, the competencies were categorizing into four domains titled specific, distinct, managerial and behavioral.

b) Based on the Delphi results, the competencies were categorized into four domains titled specific, distinct, managerial and behavioral

Apart from the error of "categorizing", it is yet to be explaied where the 19 domains come from (based on the 150) or where the difference of these sentences lies. The grouping of domains and categorizing are (in my understanding) two different steps. Figure 2 is unfortunately of no help in understanding the process. I recommend to delete the figure and replace it by a) a better flow of competencies or b) extended explanation in text.

Thank you for your valuable feedback and for highlighting areas (turquoise color) in need of further clarification. We have revised the manuscript to better explain the process of categorizing competencies from the initial 150 to the final 15, as well as the development of the four overarching domains: specific, distinct, managerial, and behavioral. Below, we address the specific points you raised:

1. Clarification of Process Steps: We have provided an expanded explanation in the methodology section to clarify the process of refining and categorizing the competencies:

o Initial Extraction and Reduction: After identifying 150 competencies, an initial review was conducted by the expert panel. Each panel member independently analyzed the competencies and suggested reductions and merges, which were then compiled and presented in a series of meetings for further refinement. This iterative review process helped us reduce the list to a manageable set of competencies.

o Categorization Framework Development: Following the reduction process, we identified the need for a structured categorization framework. After reviewing several competency models, we chose the Bartram 8G Model, which encompasses competency domains such as Analysis & Interpretation, Interaction & Influence, Support & Cooperation, Decision Making, Action & Execution, Adaptation & Adjustment, Organization, and Creation & Conceptualization. This model provided a solid foundation for organizing our competencies.

o Final Categorization and Customization: We consolidated similar competencies within the 8G framework and refined them into four key domains: specific, distinct, managerial, and behavioral. Each domain was defined based on the thematic commonalities among competencies. These final categorizations were reviewed and endorsed by the expert panel in a subsequent session.

2. Revision of Figure 2: We have updated Figure 2 to present a clearer step-by-step flowchart illustrating the categorization process, showing how the competencies were refined from 150 to 15 and organized into four domains.

We appreciate your guidance, which has helped us enhance the clarity and flow of our methodology. Please let us know if further clarification is needed.

---

## [Decision Letter · Decision Letter 3]

17 Feb 2025

Identifying Core Leadership Competencies to Success Non-Communicable Disease Control and Prevention Programs: A Mixed- Methods Study

PONE-D-23-29501R3

Dear Dr. partovi,

We’re pleased to inform you that your manuscript has been judged scientifically suitable for publication and will be formally accepted for publication once it meets all outstanding technical requirements.

Kind regards,

Dorothy Lall

Academic Editor

PLOS ONE

Additional Editor Comments (optional):

Reviewers' comments:

Reviewer's Responses to Questions

**Comments to the Author**

1. If the authors have adequately addressed your comments raised in a previous round of review and you feel that this manuscript is now acceptable for publication, you may indicate that here to bypass the “Comments to the Author” section, enter your conflict of interest statement in the “Confidential to Editor” section, and submit your "Accept" recommendation.

Reviewer #2: All comments have been addressed

2. Is the manuscript technically sound, and do the data support the conclusions?

Reviewer #2: Yes

3. Has the statistical analysis been performed appropriately and rigorously? 

Reviewer #2: Yes

4. Have the authors made all data underlying the findings in their manuscript fully available?

Reviewer #2: Yes

5. Is the manuscript presented in an intelligible fashion and written in standard English?

Reviewer #2: Yes

6. Review Comments to the Author

Reviewer #2: All corrections have been addressed. I thank the authors for making these corrections and look forward to the publication.

7. PLOS authors have the option to publish the peer review history of their article (what does this mean? ). If published, this will include your full peer review and any attached files.

**Do you want your identity to be public for this peer review?** For information about this choice, including consent withdrawal, please see our Privacy Policy .

Reviewer #2: No

---

## [Editor Report · Acceptance letter]

PONE-D-23-29501R3

PLOS ONE

Dear Dr. partovi,

I'm pleased to inform you that your manuscript has been deemed suitable for publication in PLOS ONE. Congratulations! Your manuscript is now being handed over to our production team.

Kind regards,

on behalf of

Dr. Dorothy Lall

Academic Editor

PLOS ONE